# Identification of New Substrates and Inhibitors of Human CYP2A7

**DOI:** 10.3390/molecules29102191

**Published:** 2024-05-08

**Authors:** Rana Azeem Ashraf, Sijie Liu, Clemens Alexander Wolf, Gerhard Wolber, Matthias Bureik

**Affiliations:** 1School of Pharmaceutical Science and Technology, Faculty of Medicine, Tianjin University, Tianjin 300072, China; azeem_rana@yahoo.com; 2Pharmaceutical and Medicinal Chemistry (Computer-Aided Drug Design), Institute of Pharmacy, Free University Berlin, 14195 Berlin, Germany; sijie.liu@fu-berlin.de (S.L.); ca.wolf@fu-berlin.de (C.A.W.); gerhard.wolber@fu-berlin.de (G.W.)

**Keywords:** CYP2A6, CYP2A7, coumarin, diclofenac, enzyme bags, 7-ethoxycoumarin, fission yeast, 7-hydroxycoumarin, nicotine, polymorphism, proluciferin, *Schizosaccharomyces pombe*

## Abstract

CYP2A7 is one of the most understudied human cytochrome P450 enzymes and its contributions to either drug metabolism or endogenous biosynthesis pathways are not understood, as its only known enzymatic activities are the conversions of two proluciferin probe substrates. In addition, the CYP2A7 gene contains four single-nucleotide polymorphisms (SNPs) that cause missense mutations and have minor allele frequencies (MAFs) above 0.5. This means that the resulting amino acid changes occur in the majority of humans. In a previous study, we employed the reference standard sequence (called CYP2A7*1 in P450 nomenclature). For the present study, we created another CYP2A7 sequence that contains all four amino acid changes (Cys311, Glu169, Gly479, and Arg274) and labeled it CYP2A7-WT. Thus, it was the aim of this study to identify new substrates and inhibitors of CYP2A7 and to compare the properties of CYP2A7-WT with CYP2A7*1. We found several new proluciferin probe substrates for both enzyme variants (we also performed in silico studies to understand the activity difference between CYP2A7-WT and CYP2A7*1 on specific substrates), and we show that while they do not act on the standard CYP2A6 substrates nicotine, coumarin, or 7-ethoxycoumarin, both can hydroxylate diclofenac (as can CYP2A6). Moreover, we found ketoconazole, 1-benzylimidazole, and letrozole to be CYP2A7 inhibitors.

## 1. Introduction

Cytochrome P450 enzymes (CYPs or P450s) are a large family of enzymes present in all biological kingdoms. They are hemoproteins and most famous for catalyzing monooxygenase reactions, even though many of them display a wide range of additional activities [1]. In humans, there are 57 functional P450s [2]; some of these play a role in Phase I drug metabolism and others are involved in the homeostasis of key regulator molecules such as fatty acids, vitamin D, steroids, and bile acids [3]. A number of CYPs are also involved in pathophysiological pathways and, hence, are (or might become) therapeutically relevant. Our current understanding of human CYPs is very biased, with some of them having been heavily investigated and others being understudied [4]. One of the latter enzymes is CYP2A7, which shares high similarity with the much better studied CYP2A6. CYP2A6 and CYP2A7 are both localized in the endoplasmic reticulum; for their activity, they depend on the electron transfer protein cytochrome P450 reductase (CPR). CYP2A6 plays an important role in the oxidation of nicotine and is also involved in the metabolism of other xenobiotics such as drugs, carcinogens, and alkaloids [5]. It is the only human enzyme known to efficiently catalyze the 7-hydroxylation of coumarin, and consequently this reaction is used as a probe for CYP2A6 activity [6]. Upon recombinant expression in the fission yeast *Schizosaccharomyces pombe*, we have previously shown that CYP2A7 can metabolize the two probe substrates Luciferin-H and Luciferin-ME (Appendix A) [2]. This was the first report on enzymatic activity for this P450, and to date also the only one. Therefore, it was one of the aims of this study to expand our knowledge by identifying new substrates and inhibitors of CYP2A7.

Human P450s are known to be reduced by their electron transfer partners in the absence of substrate, and in turn to reduce other acceptor molecules such as molecular oxygen, thereby creating superoxide anions (O^2−•^). This process is known as futile cycling. In a systematic screening of all 50 microsomal human CYPs, we have recently shown that both CYP2A6 and CYP2A7 have significant futile cycling activity [7]. Moreover, they were much more similar in this regard as compared to CYP2A13, the third human member of the CYP2A subfamily. This finding further indicates that CYP2A6 is likely the well-studied human P450 whose properties are most closely related to CYP2A7. For this reason, we chose to test CYP2A6 activity in parallel to all CYP2A7 reactions in this study.

Polymorphisms in the human P450 genes results in many different phenotype variants among individuals in a given population, which can cause metabolic differences [8]. CYP2A7 is one of the few human P450s for which the reference standard sequence (called *1-allele in P450 nomenclature) contains a number of side-chains that are not the most common ones in a given position [9]. In other words, there are four single-nucleotide polymorphisms (SNPs) in the CYP2A7 gene that cause missense mutations and have minor allele frequencies (MAFs) of more than 0.5 (i.e., more than 50%) (Table 1); this means that the resulting amino acid changes actually occur in the majority of humans. In our previous study mentioned above, we employed the CYP2A7*1 sequence. For the present study we created another CYP2A7 sequence that contains all four amino acid changes and labeled it CYP2A7-WT (as there is no *-number for this allele). Consequently, the second main aim of this study was to test whether CYP2A7-WT is an active enzyme and, if so, how its properties compare to CYP2A7*1.

Luciferin-based probe substrates for P450s are derivatives of firefly luciferin, which as such, are not luciferase substrates, but can be converted to d-luciferin in a CYP-dependent manner [10]. In most of these substrates, luciferin is modified by a moiety that is attached to the 6′-carbon via an ether linkage. Upon *O*-dealkylation, the 6′-hydroxy group is free, and the product can react with luciferase. In addition, the aromatic hydroxylation of a proluciferin by a CYP enzyme can also provide the 6′-hydroxy group to a substrate that lacks it. Some of these compounds (preproluciferins) need an additional reaction step for conversion into luciferin proper. In recent years we have significantly expanded the set of proluciferin probe substrates in several studies [11,12,13]. Until the present study, Luciferin-H and Luciferin-ME were the only known CYP2A7 substrates, and testing additional proluciferin probes was an obvious choice to start the current investigation.

## 2. Results and Discussion

### 2.1. Cloning of a Fission Yeast Expression Strain for CYP2A7-WT

We have previously shown that CYP2A7*1 shows activity towards the two luciferin-based probe substrates Luciferin-H and Luciferin-ME [2]. However, as explained above, at four positions (169, 274, 311, and 479), the amino acid sequence of CYP2A7*1 contains side chains that are not the most common ones [9]. Therefore, we cloned a new CYP2A7 gene sequence (called CYP2A7-WT) into the fission yeast expression vector pREP1 that contains these four changes, and used the resulting plasmid pREP1-CYP2A7-WT to transform fission yeast strain CAD62 (which contains an expression cassette for human CPR). This transformation yielded the new strain AA01 (all strains and their genotypes are listed in Table 2). All subsequent experiments were performed using both strains RAJ127 (coexpressing CYP2A7*1 and CPR) and AA01 (coexpressing CYP2A7-WT and CPR), together with strain RAJ122 (coexpressing CYP2A6*1 and CPR) for comparison, and the parental strain CAD62 (expressing CPR only) as negative control.

### 2.2. Activities of CYP2A6*1, CYP2A7*1 and CYP2A7-WT towards Different Luminogenic Substrates

As mentioned above, we have reported that CYP2A7*1 can metabolize both Luciferin-H and Luciferin-ME, albeit at comparatively low activity [2]. In this study we wanted to test the ability of this enzyme to act on more luciferin-based probe substrates and, moreover, to compare the activities of the two enzyme variants CYP2A7*1 and CYP2A7-WT. For this purpose, the following set of seven proluciferin compounds (structures are shown in Appendix A) was employed: Luciferin-H lacks the 6′-hydroxy function of the benzothiazole group of firefly luciferin; some CYP enzymes can catalyze the aromatic hydroxylation reaction that converts it into luciferin [10]. In Luciferin-ME, the 6′-hydroxy function has been derivatized to a methyl ether group (hence the name); it is a broad-range P450 substrate that can be metabolized by 29 of the 57 human CYP enzymes [2]. Luciferin-1A2 is the C-2 cyanobenzothiazole ether variant of Luciferin-ME and a known CYP2A6 substrate [14]; its reaction product is a luciferin precursor that requires an additional chemical reaction step (a coupling with d-cysteine) to form luciferin. Luciferin-BE is the 6′-benzyl ether of luciferin and a weak CYP2A6 substrate [10], while Luciferin-2FBE, Luciferin-3FBE, and Luciferin-4FBE are the corresponding luciferin-6′-monofluorobenzyl ethers with a fluorine in the ortho, meta or para position, respectively [12]. All four of these compounds are rather non-selective P450 substrates.

When testing the first three of the proluciferin substrates, it was confirmed again that CYP2A6*1 and CYP2A7*1 can both metabolize Luciferin-H and Luciferin-ME (Figure 1), as reported previously [2]. CYP2A7-WT also acted on both substrates, and while its activity towards Luciferin-H was similar to that of CYP2A7*1, its activity towards Luciferin-ME was significantly higher. For Luciferin-1A2, CYP2A6*1 activity was the highest by far, while a little bit of activity was also seen with CYP2A7*1, but not CYP2A7-WT. These results demonstrate that both CYP2A7*1 and CYP2A7-WT are active enzyme variants, and either one can catalyze aromatic as well as aliphatic hydroxylation reactions.

Biotransformations of Luciferin-BE and its monofluoro derivatives showed that all four of these compounds can be metabolized by CYP2A6*1, while two of them (Luciferin-BE and Luciferin-3FBE) are CYP2A7*1 substrates, and only Luciferin-3FBE can be converted by CYP2A7-WT (Figure 2). Taken together, these data illustrate that for some proluciferin substrates, the CYP2A6 and CYP2A7 activities are more or less similar, while for others they are not. Moreover, for one substrate (Luciferin-ME), CYP2A7-WT was the more active CYP2A7 enzyme variant, while for two others (Luciferin-BE and Luciferin-3FBE), CYP2A7*1 activities were higher.

### 2.3. Insights from Molecular Modeling into Differences between CYP2A7-WT and CYP2A7*1

In order to understand how the four amino acids that differ between CYP2A7-WT and CYP2A7*1 (Cys311Arg, Glu169Asp, Gly479Val, and Arg274His) influence the activities of these CYP2A7 variants, we explored the structures of both proteins with in silico methods. We used AlphaFold2 to predict the structures of both CYP2A7*1 and CYP2A7-WT due to the lack of publicly available high-resolution 3D atomistic structures (Figure 3). The structures were then refined and validated using the Ramachandran plot generated by the MOE Geometry phi-psi plot function (see Appendix A). The binding sites of both CYP2A7-WT and CYP2A7*1 variants are composed of Phe107, Phe111, Ala117, Phe118, Phe209, Glu245, Asn297, Ile300, Ala301, Thr305, Ile366, Leu370, Phe480, and heme, the prosthetic group. The four differing residues between the variants, Cys311Arg, Glu169Asp, Gly479Val, and Arg274His, are located far from the main binding site. Therefore, it is unlikely that these residues could directly interact with the ligands and influence ligand binding.

We aimed to illustrate the binding modes of the luminogenic substrates in their respective most efficiently metabolizing CYP2A7 variants. Therefore, we docked Luciferin-ME and Luciferin-3FBE into the binding pockets of CYP2A7-WT and CYP2A7*1, respectively. The plausible binding mode shows the molecules with their respective sites of metabolism (SoM) located sufficiently close to the heme iron. According to a well-established rule of thumb [15,16], the SoM should be located no more than 6 Å from the heme iron to facilitate substantial oxidation (Figure 4).

However, tracking the influences of these four amino acid substitutions on the protein activity for different substrates from static models that share a similar binding site alone is challenging. To account for the dynamic changes of the protein binding site, particularly the narrow area close to the heme, we performed molecular dynamics (MD) simulations with a length of 200 × 3 ns. As we hypothesized that pocket size may play a role in the preferential metabolism of Luciferin-ME and Luciferin-3FBE, we used POVME3 [17] to measure the overall volumes of the binding sites. Our results show that these volumes are nearly identical, both being around 470 Å^3^. The cross-sectional area of the tube-like pocket near the heme iron was estimated since no significant differences were found in volume measurements. A narrow bottleneck, consisting of Ile366 (Cδ), Ala301 (Cα), and Thr305 (Cγ) atoms, was identified as limiting access to the heme iron as the oxidative moiety. The cross-sectional area of the binding pocket was approximated by monitoring the area of the triangle defined by the three atoms during the MD simulations. The bottleneck of the binding site had a mean area of 10.5 Å^2^ in CYP2A7-WT and 12.2 Å^2^ in CYP2A7*1 throughout the MD simulations. The bottleneck regions of the CYP2A7-WT and CYP2A7*1 binding sites have maximum areas of 17.4 Å^2^ and 24.6 Å^2^, respectively. This suggests that the residues constituting the bottleneck in CYP2A7*1 may have a more flexible arrangement. In order to position the SoM appropriately for oxidation, the bulky benzyl group attached to the ether oxygen atom of Luciferin-3FBE must pass through the bottleneck area. This process may be easier in a more flexible binding site, such as in CYP2A7*1. It is suggested that Luciferin-3FBE is metabolized at a higher rate by CYP2A7*1 than by CYP2A7-WT based on our wet-lab data. On the other hand, a narrower bottleneck may be more favorable for stabilizing proluciferins with a smaller substituent attached to the ether oxygen, such as the SoM of Luciferin-ME, which could explain why CYP2A7-WT more efficiently oxidizes Luciferin-ME.

### 2.4. Lack of CYP2A7 Activity towards Three Known CYP2A6 Substrates

We next tested the two CYP2A7 variants for the metabolization of three standard CYP2A6 substrates: nicotine, coumarin, and 7-ethoxycoumarin. Nicotine metabolism is complex, but the main route is CYP2A6-dependent oxidation to the nicotine iminium ion (see Appendix A), which in turn is converted by aldehyde oxidase to cotinine [18].

Biotransformation experiments in this study confirmed the metabolization of nicotine (HRMS (ESI+) C_10_H_14_N_2_, [M + H]^+^ theor. = 163.1222, [M + H]^+^ exp. = 163.1223, Δ*m*/*z* = 0.6 ppm) to the nicotine iminium ion (HRMS (ESI+) C_10_H_13_N_2_, [M]^+^ theor. = 161.1079, [M]^+^ exp. = 161.1086, Δ*m*/*z* = 4.3 ppm) by CYP2A6*1, as expected. By contrast, no nicotine metabolite was found in biotransformations performed with either CYP2A7*1 or CYP2A7-WT (Appendix A). Coumarin and 7-ethoxycoumarin are two other known CYP2A6 substrates, and they can be analyzed fluorometrically [19]. Again, our biotransformation experiments confirmed the activity of CYP2A6*1 towards both substrates, while no reaction was observed for either CYP2A7 variant (Figure 5).

### 2.5. Metabolism of Diclofenac by CYP2A6 and CYP2A7

The next hint for a drug substrate of CYP2A7 came from an unexpected source: In a concurrent project (manuscript in preparation) that is aimed at testing all human CYPs for the metabolization of diclofenac (see Appendix A), which is a well-known CYP2C9 substrate [6], both CYP2A6 and CYP2A7 were found to be active. More specifically, CYP2A6*1, CYP2A7*1 and CYP2A7-WT produced a diclofenac metabolite that had the mass of hydroxydiclofenac (Appendix A). While the confirmation of the position of the hydroxy group in these diclofenac metabolites is still pending, our results demonstrate for the first time that CYP2A6*1 as well as both CYP2A7 variants can metabolize diclofenac. Moreover, this is the first evidence that CYP2A7 is indeed a drug-metabolizing enzyme. However, more studies are needed to further elucidate its substrate spectrum as well as the physiological relevance of its activity.

### 2.6. Identification of CYP2A7 Inhibitors

Having thus established functional activity assays of both CYP2A7 variants, we next set up an one-point inhibitor assay and used it to test the inhibitory potency of three standard CYP inhibitors: ketoconazole [20], 1-benzylimidazole [21], and letrozole [22]. Two of these (ketoconazole and letrozole) are already known to inhibit CYP2A6 [23]. In this assay, the substrate Luciferin-H was used at a final concentration of 100 µM, while the test compounds were used at a final concentration of 10 µM. It was found that all three CYP inhibitors significantly inhibited the activities of all three enzymes tested, with % inhibition values ranging from 58% to 80% (Table 3). No clear pattern or preference for any one enzyme was observed. These data confirm the previously described inhibition effects on CYP2A6 and, with due caution, indicate that the inhibitor spectra of CYP2A6 and CYP2A7 may overlap to some degree. Future research will show to what extent CYP2A7 contributes to drug metabolism in humans, and whether the inhibition of this enzyme might lead to drug–drug interactions. In addition, it is highly possible that this enzyme might also play a role in endogenous biosynthetic pathways.

## 3. Materials and Methods

### 3.1. Materials

Luciferin-H, Luciferin-ME, Luciferin-1A2, and the NADPH regeneration system were all obtained from Promega (Madison, WI, USA); diclofenac and DMSO were from Sigma Aldrich (St. Louis, MO, USA); nicotine was from Xi’an Taima Biological Engineering (Xi’an, China); Luciferin-BE, Luciferin-2FBE, Luciferin-3FBE, and Luciferin-4FBE were synthesized in-house using a previously published protocol [12]; 1-benzylimidazole was from Alfa Aesar (Shanghai, China); ketoconazole and letrozole were from Dingguo Biotechnology (Beijing, China); Triton-X100 was from Leagene (Beijing, China); glycerol was from Dingguo (Tianjin, China); Tris-HCl was from AKZ-Biotech (Tianjin, China); potassium chloride, ammonium bicarbonate, potassium dihydrogen phosphate, and potassium hydrogen phosphate were from Jiangtian Chemical (Tianjin, China); ethyl acetate of analytical grade was from Yuanli Chemical (Tianjin, China); and white 96-well microtiter plates were from Nunc (Thermo Fisher Scientific, Langenselbold, Germany). All other chemicals and reagents used were of the highest grade available.

### 3.2. Fission Yeast Media and General Techniques

General DNA manipulation methods were performed using standard techniques [24] and the preparation of media and basic manipulation methods of *S. pombe* were carried out as described [25]. Briefly, strains were generally cultivated at 30 °C in Edinburgh Minimal Medium (EMM) with supplements of 0.1 g/L final concentration as required. Liquid cultures were continuously shaken at 150 rpm. Thiamine was used at a concentration of 5 µM throughout.

### 3.3. Construction of Expression Plasmids and Fission Yeast Strains

The vector pREP1 [26] was used for the expression of human CYPs in fission yeast. The construction of strains RAJ122 (coexpressing human CPR and CYP2A6*1) and RAJ127 (coexpressing human CPR and CYP2A7*1) has been described before [2]. The DNA sequences for human CYP2A7-WT were synthesized by General Biosystem (Hefei, China) and cloned into pREP1 to yield pREP1-CYP2A7-WT. This vector was then transformed into strain CAD62 (expressing human CPR) [27] to yield the new strain AA01 (coexpressing human CPR and CYP2A7-WT). All strains used in this study are listed in Table 4.

### 3.4. Biotransformation with Enzyme Bags

This was performed as described previously [21] with slight modifications. Briefly, fission yeast cells were cultured on EMM plates with 5 µM thiamine for 3 days at 30 °C. A single yeast colony was then transferred to a 10 mL liquid culture of EMM without thiamine and cultured at 30 °C and 230 rpm for 36 h. As the recombinant expression of the human enzymes is controlled by the strong endogenous nmt1 promoter [28], all subsequent cultures were cultured in the absence of thiamine to keep the promoter in an active state. For each activity assay, 5 × 10^7^ cells (stationary growth phase) were transferred to 1.5 mL Eppendorf tubes (Axygen; Silicon Valley, USA), pelleted, and incubated in 1 mL of 0.3% Triton-X100 in Tris-KCl buffer (200 mM KCl, 100 mM Tris-HCl pH 7.8) at room temperature for 60 min at 150 rpm for permeabilization. After three washes with 1 mL NH_4_HCO_3_ buffer (50 mM, pH 7.8), cells were resuspended in phosphate buffer (pH 7.4) and then the substrate and the NADPH regeneration system were added to a total volume of 50 µL. The samples were incubated at 37 °C and 1000 rpm for 3 h. For LC-MS analysis, the reaction mixture was extracted with an equal amount of ethyl acetate by centrifugation, and the supernatants were stored at −20 °C until use.

### 3.5. Bioluminescence Detection

This was performed as described previously [21] with slight modifications. Briefly, a concentrated CYP reaction mixture (containing a fourfold concentrated substrate and potassium phosphate buffer) was added to the cell pellets after the permeabilization and washing process. For inhibition assays, the CYP reaction mixtures with the inhibitors at a final concentration of 10 µM were pre-incubated at 37 °C for 10 min. CYP reactions were commenced by adding the twofold concentrated NADPH regeneration system. Samples were incubated for 3 h at 37 °C and 1000 rpm. After centrifugation at 16,000× *g* for 1 min, the supernatants were transferred to the white microtiter plates and an equal amount of reconstituted luciferin detection reagent was added to each well. Plates were then incubated at room temperature for 20 min and the luminescence was recorded on a Magellan infinite 200Pro microplate reader (Tecan; Männedorf, Switzerland). For the proluciferin substrates’ screening, the specific reaction conditions and substrate concentrations were as given in the instructions of the manufacturer (Promega). In all cases, reaction parameters (reaction times and enzyme concentrations) were within the linear range. For inhibitor assays, ketoconazole, 1-benzylimidazole, and letrozole were added to a final concentration of 10 µM. All measurements were done at least three times in triplicates.

### 3.6. Fluorescence Detection of Coumarin 7-Hydroxylation and Deethylation of 7-Ethoxycoumarin

The fluorescence activity of CYP enzymes was measured as described [19] with slight modifications. The solution was shaken at 37 °C for 3 h in the dark at a 50 µL final reaction volume. Afterwards, reactions were stopped by adding 350 µL ethyl acetate. The reaction tubes were vortexed for 5 min before centrifugation at 16,000× *g* for 8 min. Finally, 200 µL of the upper phase was extracted three times with 350 µL of ethyl acetate each. In the last step, the reaction tubes were air-dried in the dark. For reconstituting the pellet we added 50 µL of methanol, which was later followed by the addition of 250 µL of 0.01 M NaOH. The amounts of de-ethylated products were quantified by using standard solutions of 7-hydroxycoumarin and 7-ethoxycoumarin. The concentrations of 7-hydroxycoumarin were determined fluorometrically with 50 µL volume using Grenier microlon flat-bottom 96-well black plates. The energies for excitation and emission were set at 368 and at 530 nm, respectively. All readings were taken in triplicate.

### 3.7. Analysis of Nicotine Metabolites

An Agilent HPLC/microTOF-Q II instrument equipped with a LiChrospher 100 RP-18 column was used to carry out all analytical procedures. The mobile phase that we used was a mixture of water with 0.05% formic acid as phase A and 100% acetonitrile as B, run at a constant flow of 0.3 mL/min for 45 min. The gradient used was 0–20 min (80% A: 20% B to 80% B), 20–30 min (100% B), 30–35 min again (100% B), and 35–45 min (80% A: 20% B). We used positive electrospray ionization mode at a capillary voltage of 4.5 kV for mass spectrometry. Moreover, we set a gas flow of 4 L/min at 190 °C, and for nebulizer pressure, we used a pressure of 0.8 bars. The final concentration of nicotine was 250 µM and the UV detector was set at 260 nm.

### 3.8. Analysis of Hydroxydiclofenac on HPLC/microTOF-QII

Analytical procedures were performed on an Agilent HPLC/microTOF-Q II instrument equipped with a Kromasil 100-5-C18 (dimensions 4.6 × 250 mm). The phase was 100% water (phase A) and 100% methanol (phase B), run at a constant flow of 0.3 mL per min for 50 min. The gradient used was 0–15 min (90% A: 10% B to 30% B), 15–30 min (50% B), 30–45 min again (70% B), 45–60 min (10% A: 90% B), and 60–75 min (10% A: 90% B). The MS system was operated in negative electrospray ionization mode at a capillary voltage of 4.5 kV. A drying gas flow of 4 L min/min at 190 °C and a nebulizer pressure of 0.8 bar were used. The final concentration of diclofenac was 100 µM and the UV detector was set to 280 nm for the detection of diclofenac and hydroxydiclofenac. An injection volume of 20 µL was used.

### 3.9. Analysis of Hydroxydiclofenac on HPLC/micro-QE

Analytical procedures were performed on an Agilent HPLC/micro-QE instrument equipped with a ZORBAX SB-C18 column (dimensions 4.6 × 150 mm). The phase was 100% water (phase A) and 100% methanol (phase B), and run at a constant flow of 0.5 mL per min for 35 min. The gradient used was 0–1 min (90% A: 10% B), 2–6 min (30% B), 7–11 min again (50% B), 12–16 min (70% B), and 17–35 min (90% B). The MS system was operated in negative electrospray ionization mode at a capillary voltage of 4.5 kV. A drying gas flow of 4 L min/min at 190 °C and a nebulizer pressure of 0.8 bar were used. The final concentration of diclofenac was 100 µM and the UV detector was set to 280 nm for the detection of diclofenac and hydroxydiclofenac. An injection volume of 20 µL was used.

### 3.10. Statistical Analysis

All data are presented as mean ± SD. Statistical significance was determined using a two-tailed *t*-test. Differences were considered significant if *p* < 0.05. Statistical analysis was performed using GraphPad PRISM 5. 01 (GraphPad Software, Inc., La Jolla, CA, USA).

### 3.11. Protein Modeling and Molecular Dynamics (MD) Simulations

Atomistic models for both CYP2A7*1 and CYP2A7-WT were generated using AlphaFold v. 2.1.0 [29]. The heme prosthetic group was inserted into the structure of CYP2A7*1 and CYP2A7-WT, copied from the most similar crystal structure of CYP2A6 (PDB code: 2FDU.A [30]). The apo structures containing the prosthetic group heme were prepared in MOE v. 2020.0901 (Molecular Operating Environment; Chemical Computing Group ULC, Montreal, QC, Canada) using the integrated Structure Preparation and Protein Builder tools to optimize the structure, and Protonate3D [31] with the OPLS-AA force field [32] parameters.

Each prepared model was loaded into Maestro v. 13.1.137 (Schrödinger Release 2022-1: Maestro, Schrödinger, LLC, New York, NY, USA) to perform all-atom MD simulations. The structure and environment were prepared with the implemented functionalities “Protein Preparation Wizard” and “System Builder”. Termini were capped and missing disulfide bonds were automatically added. Both proteins’ apo structures were individually solvated in cubic water boxes with padding of 10 Å filled with TIP4P water model [33]. One chloride ion was added to each system for neutralization, and another 0.15 M NaCl was added to mimic the physiological solvent. The generated systems were simulated with the Desmond simulation engine v. 6.9 [34] on water-cooled Nvidia RTX 2080 Ti graphical processing units (GPUs) for 200 ns in three replicates for each system, using the OPLS-AA force field. The simulation temperature and pressure were kept at their default values of 300 K and 1.01325 bar using the Nose–Hoover chain thermostat method and the Martyna–Tobias–Klein barostat method, respectively. The systems were relaxed and equilibrated following the standard seven-step protocol. System coordinates were recorded every 200 ps, so that in each replica of 200 ns simulation, 1000 frames were saved as trajectory files. The coordinate and trajectory files were wrapped and aligned with VMD v. 1.9.3 [35]. Bonds and angles formed by amino acid residues of interest were monitored and recorded. The area of the triangle defined by two adjacent bond lengths A and B and the angle α between them was calculated for each frame using Python (version 3.11.0) according to the formula: area [Å^2^] = 1/2 × length of bond A [Å] × length of bond B [Å] × sin(α). Key libraries included NumPy (version 1.26.0) and Pandas (version 2.1.3), used for their robust capabilities in numerical and data frame manipulation, respectively.

### 3.12. Molecular Docking

Luciferin-ME was docked into the binding pocket of CYP2A7-WT and Luciferin-3FBE was docked into the binding site of CYP2A7*1 using GOLD v. 5.8.1 (Genetic Optimisation for Ligand Docking; CCDC Software, Cambridge, UK) [36]. In total, 20 genetic algorithm (GA) runs with the coordinates of the heme iron serving as the operation center were used with a radius of 10 Å. The operation was directed at generating diverse solutions with early termination prevented. A 100% search efficiency was used and pyramidal nitrogen atoms were allowed to flip. In order to ensure substrate conformations close to the heme iron, the distance between the site of metabolism (SoM) of each substrate and the heme iron was constrained to between 1.5 and 5 Å, with a spring constant of 5.0. In the case of Luciferin-ME, the SoM was the terminal carbon atom attached to the ether oxygen, while for Luciferin-3FBE, the SoM was the benzylic carbon atom attached to the ether oxygen. Other than that, default settings were maintained, and the resulting conformations were analyzed using Ligandscout v. 4.4.3 [37,38] and minimized using the implemented MMFF94 force field [39]. Reasonable poses were selected based on the distance between SoM and heme iron as well as stereochemical plausibility.

## 4. Conclusions

In our previous work, we demonstrated for the first time that the CYP2A7 gene codes for an active enzyme. In the present study, we have expanded the CYP2A7 substrate spectrum and compared the activities of the two enzyme variants CYP2A7*1 and CYP2A7-WT, which differ at four positions in their amino acid sequence. Their enzymatic properties were found to be similar, but not identical. We also demonstrated that diclofenac is a substrate for CYP2A7 (as well as CYP2A6), and identified the first three inhibitors for this enzyme, which are ketoconazole, 1-benzylimidazole, and letrozole, respectively.

## Figures and Tables

**Figure 1 molecules-29-02191-f001:**
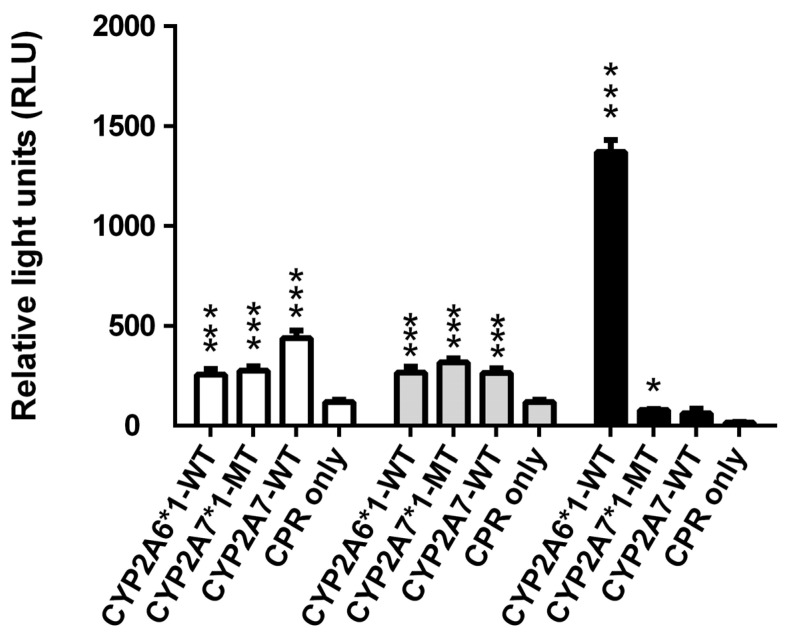
Conversion of the three luminogenic substrates Luciferin-ME (white columns), Luciferin-H (grey columns), and Luciferin-1A2 (black columns) by permeabilized fission yeast cells coexpressing CPR with either CYP2A6*1, CYP2A7*1, or CYP2A7-WT. Cells expressing CPR only were also tested (negative control). Activities are shown in relative light units (RLU). * *p* < 0.05, *** *p* < 0.001 vs. negative control.

**Figure 2 molecules-29-02191-f002:**
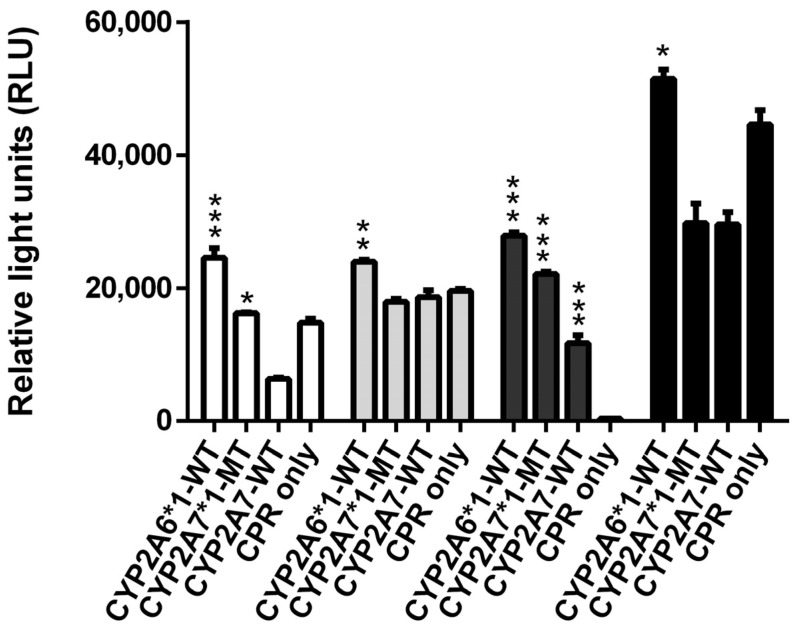
Conversion of the four luminogenic substrates Luciferin-BE (white columns), Luciferin-2FBE (light grey columns), Luciferin-3FBE (dark grey columns), and Luciferin-4FBE (black columns) by permeabilized fission yeast cells coexpressing CPR with either CYP2A6*1, CYP2A7*1, or CYP2A7-WT. Cells expressing CPR only were also tested (negative control). Activities are shown in relative light units (RLU). * *p* < 0.05, ** *p* < 0.01, *** *p* < 0.001 vs. negative control.

**Figure 3 molecules-29-02191-f003:**
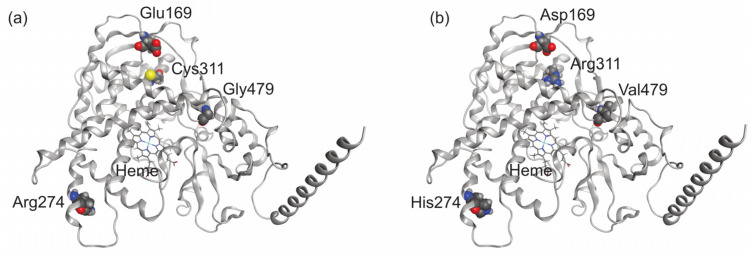
Predicted atomistic model of (**a**) CYP2A7*1 and (**b**) CYP2A7-WT. The four different residues between CYP2A7*1 and CYP2A7-WT are presented in space-filling ball-and-stick form.

**Figure 4 molecules-29-02191-f004:**
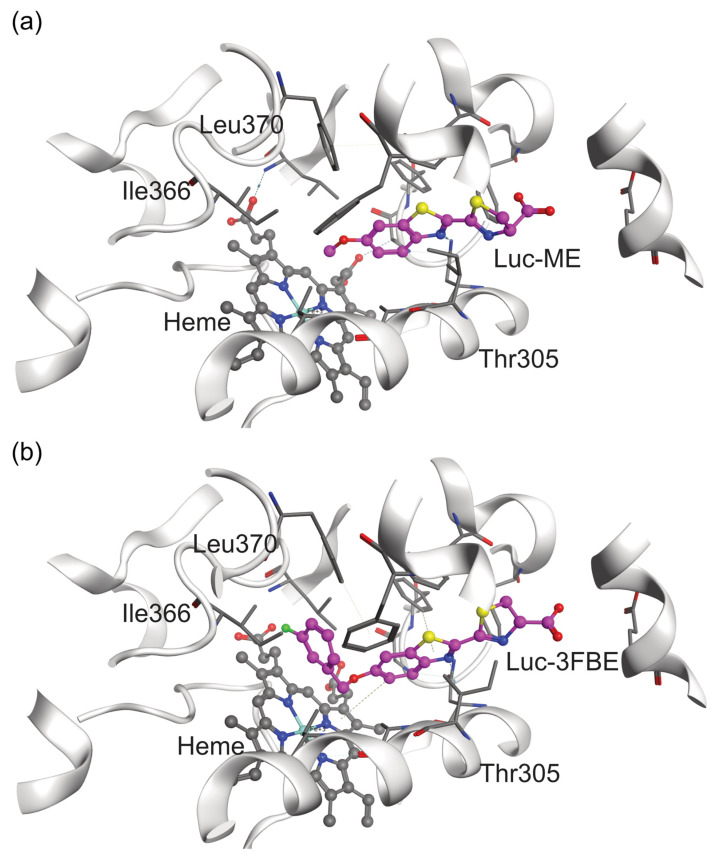
Docking conformations of (**a**) Luciferin-ME and (**b**) Luciferin-3FBE in CYP2A7-WT and CYP2A7*1, respectively. Heme is colored gray and the proluciferins are colored violet, and both are presented in ball-and-stick form.

**Figure 5 molecules-29-02191-f005:**
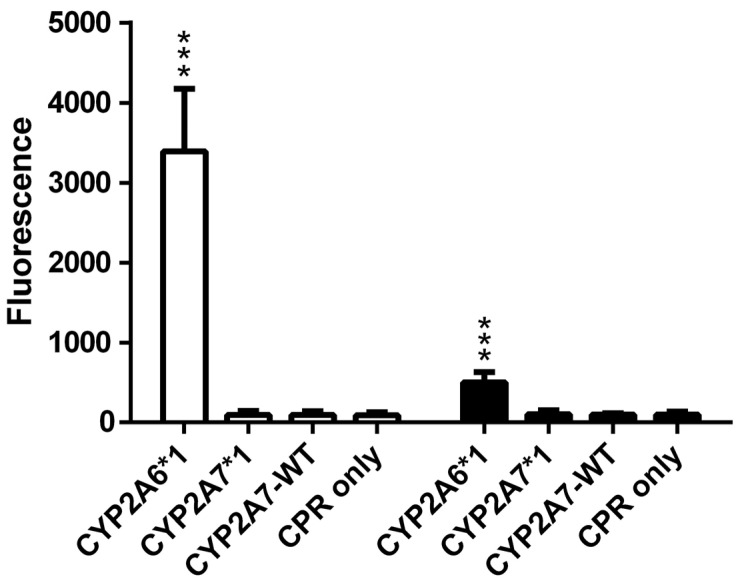
Conversion of coumarin (white columns) and 7-ethoxycoumarin (black columns) to 7-hydroxycoumarin by permeabilized fission yeast cells coexpressing CPR with either CYP2A6*1, CYP2A7*1, or CYP2A7-WT. Cells expressing CPR only were also tested (negative control). Activities are shown as relative fluorescence. *** *p* < 0.001 vs. negative control.

**Table 1 molecules-29-02191-t001:** Sequence variations in CYP2A7 where the reference allele is not the major allele.

Reference SNP ^a^	Minor Allele Frequency	Residue Change ^b^
rs3869579	0.47770050	Cys311Arg
rs4142867	0.47727665	Glu169Asp
rs12460590	0.33779851	Gly479Val
rs4079366	0.24078928	Arg274His

^a^ Single-nucleotide polymorphism. ^b^ The first amino acid is that of the WT sequence and the second is that of the *1-sequence.

**Table 2 molecules-29-02191-t002:** Activity comparison of CYP2A7 variants towards seven different luminogenic substrates.

Enzyme	Luminescence (RLU)						
	Luciferin-H	Luciferin-ME	Luciferin-1A2	Luciferin-BE	Luciferin-2FBE	Luciferin-3FBE	Luciferin-4FBE
CYP2A7*1	318 ± 60	276 ± 63	79 ± 11	16,216 ± 369	17,900 ± 1470	22,110 ± 1060	29,790 ± 8760
CYP2A7-WT	263 ± 72	439 ± 113 **	63 ± 67	6312 ± 699 *	18,610 ± 3160	11,720 ± 3560 ****	29,580 ± 5550

Significant differences: * *p* < 0.05, ** *p* < 0.01, **** *p* < 0.0001 vs. CYP2A7*1.

**Table 3 molecules-29-02191-t003:** Inhibition of CYP2A6*1, CYP2A7*1, and CYP2A7-WT by several known P450 inhibitors.

Compound	Structure	% Inhibition of CYP2A6*1	% Inhibition of CYP2A7*1	% Inhibition of CYP2A7-WT
Ketoconazole	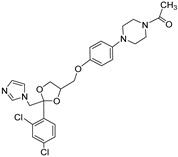	70 ± 11 ****	74 ± 4 ****	58 ± 17 ****
1-Benzyl-imidazole	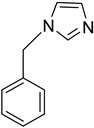	64 ± 12 ****	55 ± 11 ****	80 ± 21 ****
Letrozole	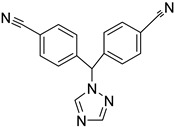	64 ± 10 ****	78 ± 7 ****	72 ± 13 ****

Inhibitor concentration: 10 µM. Substrate concentration: 100 μM. Significant differences: **** *p* < 0.0001 vs. control.

**Table 4 molecules-29-02191-t004:** Fission yeast strains used in this study.

Strain	Expressed Protein(s)	Parental Strain	Genotype	References
CAD62	CPR	NCYC2036	*h^-^ ura4-D18 leu1::pCAD1-CPR*	[27]
RAJ122	CPR, CYP2A6*1	CAD62	*h^-^ ura4-D18 leu1::pCAD1-CPR/* *pREP1-CYP2A6*1*	[2]
RAJ127	CPR, CYP2A7*1	CAD62	*h^-^ ura4-D18 leu1::pCAD1-CPR/* *pREP1-CYP2A7*1*	[2]
AA01	CPR, CYP2A7-WT	CAD62	*h*^-^ *ura4-D18 leu1::pCAD1-CPR/* *pREP1-CYP2A7-WT*	This study

## Data Availability

The raw data supporting the conclusions of this article will be made available by the authors on reasonable request.

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
