# Peer review of "Identification of New Substrates and Inhibitors of Human CYP2A7"

_molecules, 2024, doi:10.3390/molecules29102191_

Round 1
Reviewer 1 Report
Comments and Suggestions for Authors
In their paper, Ashraf et al describes the produсtion of cytochrome P450 enzyme CYP2A7 with four mutations and investigate the substrate specificity of mutant CYP2A7 and two other cytochromes. The authors expressed enzymes, determined conversion of various luciferines and anti-inflammatory drug diclofenac and studied the inhibition by several P450 inhibitors.
I recommend publication of this paper in Molecules after minor corrections have been made.
Line 21. In silico should be italic. Same for line 169.
Keywords. Schizosaccharomyces pombe should be italic. Same for line 46.
Table 2. The accuracy of some luminescence values is too high to match the accuracy of the experiment. 17902±1470 should be rounded to 17900±1470. 18608±3160 should be rounded to 18610±3160. 22107±1064 should be rounded to 22110±1060. 11722±3563 should be rounded to 11720±3560. Same for Luciferin-4FBE.
Line 240. C10H14N2. The number of atoms should be subscript. Same for lines 241 and 367.
Line 398. 7 ethoxycoumarin. The hyphen is missing.
Supplementary. Figure S2. “(ESI-QTOF-LCMS) of nicotine” should be removed.
Supplementary. Figures S9, S10, and S11. C14H10Cl2N1O3. “l” of “Cl” should not be subscript.
Reviewer 2 Report
Comments and Suggestions for Authors
In this manuscript, Bureik and coworkers reported the discovery of new substrates and inhibitors of human CYP2A7. Due to the high significance of cytochrome P450 enzyme in drug metabolism, discovery and development of small molecule inhibitors against CYPs are in high demand. However, selective inhibition represents a huge challenge to the medicinal chemistry community because of the highly conserved structures. The authors cloned a few strains and tested their activities against some substrates. However, the authors did not solve the most challenging problem of ligand promiscuity.
There are some concerns about this manuscript.
1. The citation style seems not eligible with Molecules.
2. In the introduction part, the authors introduced the background of this project, especially the substrates of CYPs. One figure should be created to illustrate the reaction process of their substrates.
3. Line 92, Line107: what is the difference between these two lines?
4. Line 112: there should be some figures to show the structures of these seven proluciferin compounds
5. In table 3, the authors conducted inhibitory assay to test the activity of these three ligands. Some values are pretty similar. To give a better understanding of the selectivity, the authors should titrate the drugs and calculate the IC50 for each compound.
6. Because the authors try to compare the CYP6, 7 and mutations, however, it is really hard for readers to follow up with the different names and their structures. The authors should use a figure to analyze the alignment of these sequences, which can be used to support the observation of low selectivity of these three drug molecules.
7. I did not understand the CPR in the context. The authors did not give any explanation of this abbreviation.
8. The authors did not test the concentration of the CYP proteins in each strain which might be the cause of different activities. A western blot should be done to compare the concentration of different proteins in different strains. Otherwise, the conclusion will be very weak. This is the main concern and must be solved!
9. Line 88: Since prior to this study
10. Line 291: extend
Comments on the Quality of English Language
1. Line 88: Since prior to this study
2. Line 291: extend
